

# Measurement of limb radiance and Trace Gases in UV over Tropical region by Balloon-Borne Instruments - Flight Validation and Initial Results

A. G. Sreejith[1], Joice Mathew[1], Mayuresh Sarpotdar[1], K. Nirmal[1], S. Ambily[1], Ajin Prakash[1], Margarita Safonova[1], and Jayant Murthy[1]

[1]Indian Institute of Astrophysics, Bangalore, India

*Correspondence to:* A. G. Sreejith (agsreejith@iiap.res.in)

**Abstract.**

We describe a method to measure the solar scattered UV radiance and trace gases in the upper troposphere and stratosphere using a spectrograph carried by high-altitude balloons. The flights have been conducted over tropical regions near urban areas. The instrumentation was developed using off-the-shelf components and can be easily reproduced. We describe the design of the
5 instrument, its operation in flight and discuss some flight result. Observations of the scattered solar radiance were performed by looking towards the Earth's limb at Sun angles of more than 90 degrees at altitudes of 10 to 29 km. This method allows us to estimate the variation of the scattered UV solar radiance with height. It also can be used to measure the column densities in the upper atmosphere.

## 1   Introduction

Balloon-borne instruments are a comparatively cheap and rapid method of measuring *in situ* profiles of ozone and other trace gases (Okano et al., 1996) in the upper troposphere and stratosphere. These have typically been done by electro-mechanical devices tuned for specific gasses (Wolff et al., 2008). Spectroscopic observations of absorption and emission lines in the atmosphere allow simultaneous measurement of multiple gas species — including trace gases — and have enhanced our knowledge of photochemical reactions of atmospheric chemistry in the atmosphere (e.g. Weidner et al., 2005; Kritten et al.,
2010).

Space-based instruments such as SCIAMACHY (Scanning Imaging Absorption Spectrometer for Atmospheric Chartography: Bovensmann et al., 1999) or SOLSE/LORE (Space Shuttle Ozone Limb Sounding Experiment/Limb Ozone Retrieval Experiment: McPeters et al., 2000) have the advantage of being able to monitor changes in the atmosphere over long periods of time. A complementary approach is to use payloads flown to altitudes of 20–40 km by high-altitude balloons (Differential
Optical Absorption Spectrometer (DOAS) instrument: Ferlemann, et al., 2000) with different observational geometries.

We have initiated a high-altitude balloon program at the Indian Institute of Astrophysics to develop low-cost instruments for use in atmospheric and astronomical studies (Nayak et al., 2013; Safonova et al., 2016). In this work, we have explored the feasibility of using an off-the-shelf compact, low-power spectrograph to trace UV absorption in the atmosphere. We describe





the instrumental set-up and calibration of the spectrograph, and the results from two flights conducted on 15th June and 12th October, 2014. We obtained the variation of the scattered solar UV (UVA/UVB) radiance, and measured the ozone slant column density (SCD) with altitude. Although we obtained valuable data using off-the-shelf instrument, we believe that a custom-built spectrograph will prove more productive.

## 2   The instrumental set-up

Our balloon-borne instrumental set-up consisted of three main parts: (i) a spectrograph with an optical fiber and a light collecting lens; (ii) a Raspberry Pi single board computer (SBC); (iii) an optical monitoring camera and environment sensors; and (iv) an attitude sensor designed and developed by Sreejith et al., (2014). All instruments were powered through a 13000 mAh lithium battery with two power settings at 5 W and 10 W. Figure 1 (*Left*) shows the schematics of the instrumental set-up. The entire payload was placed in a STYROFOAM™ (Dow Chemical Company, USA) box to keep the electronics within their operating temperatures (above $0°C$). We found that, in practice, we had to make a small opening in the payload box to allow the heat from the electronics to escape.

To validate this method, we have flown this set-up on two high-altitude balloon flights which are described in Section 3.

### 2.1   UV spectrograph

We used a CCD-based miniature holographic grating spectrograph (MayaPro2000 from Ocean Optics, Inc, USA) operating in a symmetrical crossed Czerny-Turner configuration with high sensitivity in the UV range (200–400 nm). Light is collected through a 20 mm-wide lens with a field of view (FOV) of $2°$. The collected light is fed to the spectrograph through an optical fiber ($100\,\mu$m dia) placed at the focus of the lens. The aperture of the fiber also acts as the entrance slit for the spectrograph. The light from the entrance slit is focussed onto the fixed holographic UV grating (1200 lines/mm) by the collimating mirror (see the schematic diagram in Fig. 1,*Right*). The peak efficiency of the grating is above $30\%$ in the wavelength band of 200–420 nm with a full-width half-maximum (FWHM) of 0.75 nm. The diffracted light is reflected by the grating which is focussed by the mirror onto a FFT (Full Frame Transfer) back-thinned CCD from Hamamatsu (S10420-1106-01) with a quantum efficiency of over $40\%$ in the 200–400 nm wavelength band. The spectrum is spread over the CCD array which consists of $2048 \times 64$ pixel elements. The spectrograph is controlled by an on-board microcontroller which handles the A/D conversion and the communication with the payload computer. The detailed specifications of the spectrograph are provided in Table 1.

### 2.1.1   Calibration and Testing

### 2.1.2   Dark count

We measured the dark rate of the spectrograph in a dark room with the entrance slit covered by a black cap. The temperature sensor (LM75) was placed on the surface of the spectrograph close to the detector side effectively measuring the detector temperature. We performed this test at several different temperatures in the temperature-controlled room, and found that the



dark count varies linearly with the temperature (Fig. 2, *Left*). A typical dark count signal for an exposure of 1 s is shown in Fig. 2 (*Right*), where the mean value of the counts is $2730 \pm 7$. Since solar radiation at wavelengths shorter of 250 nm does not penetrate below 40 km where our observations usually take place, any recorded counts will be only the dark counts, therefore observations at wavelengths below 250 nm can be used for the dark signal correction (Wolff et al., 2008). Figure 3 shows

the dark signal observed below 250 nm during flight (October 12, 2014) for an internal temperature of $28\,^{\circ}\mathrm{C}$ (recorded by the internal temperature sensor) and as observed in the laboratory for the same temperature. The variation of dark current with temperature followed the same trend as observed in the laboratory (Fig. 2, *Left*).

### 2.1.3 Wavelength calibration

The nominal wavelength calibration of the spectrograph from the manufacturer (Ocean Optics Manual, 2009) has a third order

relation between the pixel value and wavelength:

$$\lambda_p = I + C_1 p + C_2 p^2 + C_3 p^3 , \tag{1}$$

where $\lambda$ is the wavelength at pixel $p$, $I$ is the wavelength of 0th pixel, and $C_1$, $C_2$ and $C_3$ are the coefficients. The spectrograph comes wavelength-calibrated from the manufacturer, and the vendor provides frequent wavelength calibration upon request.

### 2.2 Attitude sensor and other monitors

We have used a compact MEMS-based attitude sensor built in-house by Sreejith et al. (2014) to monitor the viewing direction of the spectrograph. This sensor uses an internal IMU (inertial measurement unit) with an external GPS unit to give the pointing to an accuracy of $\pm 0.24^{\circ}$ in either RA (Right Ascension) and DEC (Declination), or in Earth-centered inertial coordinates (azimuth and elevation). The technical details are given in Table 2. The attitude sensor provides azimuth angle from Earth's magnetic North in clockwise direction and elevation w.r.t. the Earth's surface every 30 ms. Only those elevation angles that are

within $\pm 1^{\circ}$ are considered as limb view direction. Azimuth angles around $270^{\circ}$ were selected to ensure sun avoidance angle of more than $90^{\circ}$.

Environment sensors (temperature, humidity and pressure) are usually employed to measure atmospheric parameters during the flight. Two temperature sensors, internal (inside the payload box) and external (outside the payload box), monitor the temperature of the payload (the internal sensor is placed to ensure that the spectrograph remains within its operational temperature

range). A GPS sensor provides the positional information of the payload. The technical details of the environment sensors are given in Table 3. The horizontal camera is a follow-up instrument to verify the look direction of the light-collecting aperture. A second camera is placed vertically to monitor the balloon and records images every 30 s.



## 2.3   On-board computer

The payload is controlled through a Raspberry Pi[1] single-board computer (SBC) running a Linux operating system. We have developed the python code (MCDAS[2]) to send commands to the spectrograph and to receive the data through the USB port.

The on-board computer also controls the monitoring hardware (Section 2.2) in the payload through the Raspberry Pi's I2C

port using codes developed in-house (also available at our web-page). The data from the attitude sensor, optical cameras and the environment sensors are stored on-board and retrieved after the recovery of the payload.

## 3   Description of the experiment

The spectroscopic observation of UV scattered light was carried out on two flights in 2014, on 15th June and on 12th Oc-tober. The launches were carried out in the morning, when winds were relatively mild (Table 4), from the Hosakote campus

($13.113°$N, $77.814°$E) of the Indian Institute of Astrophysics (Nayak et al., 2013, Safonova et al., 2016), near Bangalore, In-dia. The payload, along with the instrument (Sec. 2), also included a radio telemetry unit, batteries, a parachute and two flight termination units (FTU). The light-collecting lens, attitude sensor and an optical camera were placed horizontally with respect to the base of the payload in order to look at the Earth's limb. Another optical camera was placed looking up, recording the images of balloons every 30 s for balloon monitoring. This helps us to keep a record of the time of balloons burst/cut-off, as

well as the functioning of the FTU. Power for the instruments was provided by lithium polymer batteries.

The total weight of payload including the parachute was about 5 kg in both experiments, which were launched using three sounding balloons[3]. Each balloon was filled with 7 cubic meters of hydrogen, corresponding to a measured neck lift of 5.5 kg. The ascent rate of the balloons is typically 6 m/s with a final altitude close to 30 km. Our balloons usually take around 90 minutes to reach the maximum altitude, therefore the FTU system was programmed to detach the balloon from the parachute

after 100 minutes of the flight. Typical descent rates with the parachute are about 4.5 m/s.

The ground track for the June flight was obtained from GPS data (Fig. 4) which is accurate to better than 10 meters on the ground and to about 20–30 meters in altitude. The GPS transmitted the position of the payload to the ground station at the CREST campus through a radio link operating at a frequency of 144 MHz. We monitored the altitude profile (Fig. 5, *Left*), and the variation of the temperature inside and outside the payload box (Fig. 5, *Right*). The payload experienced oscillations

in both azimuth and elevation with an average rotation of $\sim 10°$/s in azimuth and a maximum variation of $\sim \pm10°$ in elevation (Nirmal et al., 2016). The GPS radio tracker failed to provide location information for the October flight because the antenna broke 10 minutes into the flight. We have derived the altitude for this flight using the information of the general trend of ascent rates from our previous flights (Safonova et al. 2016). The time of reaching the maximum altitude was obtained from the image of the balloon burst/cut-off. However, the derived altitude information on October flight was not accurate enough to be used in

analysis of the scattered light, therefore, we used here only the data from the June flight.

---

[1]Raspberry Pi Foundation, UK

[2]Available as open source at our web-page `https://github.com/iiabaloongroup`

[3]CPR-2000, Pawan balloons, India.





We programmed the spectrograph to record spectra every 5 seconds with an exposure time of 1 second. The time-tagged data from the spectrograph are stored on board along with the housekeeping data (payload orientation and environment sensors) and images from the cameras. These are recovered after landing and analyzed in the laboratory.

## 4 Observations and Results

### 4.1 Scattered Solar Light

The aperture was pointed toward the Earth limb with the solar zenith angle (SZA: the angle between the Sun and the zenith) varying between $53°$ and $41°$. $53°$ corresponds to the SZA at launch (ground), and $41°$ corresponds to the SZA at maximum altitude. We only considered observations taken at elevation angles of $\pm 1°$ and at azimuth angles around $270°$ from magnetic North using attitude sensor data (Sec. 2.2). The angle from the Sun was further verified with the optical camera.

We have calculated the relative radiance in two UV bands: UVA (315–420 nm) and UVB (280–315 nm), and plotted them with respect to height of observation (Fig. 6). The relative radiance in these bands is given by the equation

$$I_{band} = \frac{\Sigma I_\lambda}{\Delta\lambda}, \tag{2}$$

where $I_{band}$ is the radiance in the particular band (UVA/UVB), $I_\lambda$ is the radiance at a particular wavelength and $\Delta\lambda$ is the wavelength band. Figure 6 shows the variation of measured radiance in the two different wavelength bands; the points are the measurements at different heights and the dotted lines represent an exponential fit in agreement with results obtained in Weidner et al. (2005). The payload was oscillating too much to obtain useful results below 10 km (Sreejith et al., 2014). Here we have only considered the data above 10 km – this is because the scattering below 10 km is heavily influenced by cloud cover, and the payload had too much oscillations, which has prevented us from obtaining the useful results. The SZA does not have a linear relation with altitude because our payload has lateral movement during the observations therefore, we have plotted the radiance vs SZA in a separate graph (Fig. 7).

### 4.2 Trace Atmospheric Gases

Several species of trace gases ($O_3$, BrO, $NO_2$, etc) have strong absorption features in the observed wavelength range (200–420 nm). In this paper, we present the results of the ozone column densities derived using the widely accepted Differential Optical Absorption Spectroscopy (DOAS) (Platt and Stutz, 2008) technique. The DOAS method is based on the Lambert-Beer law where it separates aerosol extinction and broadband molecular absorption from narrow-band trace gas absorption (Platt and Stutz, 2008),

$$I(\lambda) = I_0 \cdot e^{-\sigma(\lambda)\cdot S} \cdot e^{-\tau_R(z)}, \tag{3}$$

where $I$ and $I_0$ are observed and reference radiances, respectively, $S$ is Slant Column Density (SCD) which is the output of the DOAS technique. $\sigma(\lambda)$ is the absorption cross-section and is a property of the observed trace gas, available in the literature.





The optical thickness of Rayleigh scattering is given by $\tau_R(z)$ at a height $z$. We have used DOASIS (Kraus, 2004) which is a software to analyze the data and obtain the trace gas strengths. DOASIS uses a least-square fitting with cross-sections of different species and a synthetic Ring spectrum[4], generated using DOASIS software (Kraus, 2004), in the wavelength band of interest to generate the differential slant column densities (dSCD) of the trace gases. A third-order polynomial was used to account for the broadband features (Rayleigh and Mie scattering).

The observed spectra were corrected for the dark current measured on the ground before launch before the DOAS analysis. The DOAS fit was carried out in the wavelength range 305–340 nm. The following cross-sections were used: ozone from Bogumil et al. (2003) at $T = 223$ K, $NO_2$ from Vandaele et al. (1998) at $T = 220$ K, BrO from Wahner et al. 1988 at $T = 228$ K and $O_4$ at $T = 296$ K from Spectroscopy Lab of Royal Belgian Institute for Space Aeronomy (http://spectrolab.aeronomie.be/o2.htm). Though absorption cross sections depend on temperature (Chehade et al., 2013), the variation is small within our fitting window. A Fraunhofer reference spectrum (solar spectrum (Kurucz et al., 1984) convolved with the instrument's slit function), along with various trace gas absorption cross-sections and the Ring spectrum, was used for analysis. A shift-and-squeeze technique was carried out with respect to the Fraunhofer spectrum during the fitting process. The sample spectral retrieval using DOAS analysis is shown in Fig. 8.

The DOAS method only retrieves differential Slant Column Densities (dSCD), i.e. the trace gas absorption of the spectrum w.r.t. the solar reference spectrum. However, SCD of a strong absorber like ozone can be retrieved, using as the reference the solar spectrum (Kurucz et al., 1984) convolved with the instrument's resolution. We have computed the ozone SCDs at different heights using the DOAS analysis and the result from June flight is shown in Fig. 9. Because the observations were carried out during cloudy conditions, we did not use the data taken below 10 km for DOAS analysis (Section. 4.1).

However, this technique cannot be used for trace gases like $NO_2$, BrO, etc. whose optical densities are very small (Weidner et al., 2005). For these trace gases, observations at float altitude must be used as the reference, where the trace gas strength at these reference heights can be found from radiative transfer calculations. Our signal to noise ratio (SNR) of 2.06 was too low to calculate these trace gases from current observations; besides these flights were not intended for float.

## 5 Conclusions and Future Developments

Here we present a method to measure the limb scattered UV radiance and trace gas strengths using the balloon-borne UV spectrograph. The instrument discussed here have been flown on some of our balloon flights, and is in a constant state of improvement. This method for atmospheric observations provides a system which is comparable to other existing systems in the field, and its functionality is well-documented in the previous sections. The instrument used in the experiment suffers from certain limitations that have to be corrected. For example, the current spectrograph has a low signal to noise, therefore, improving the SNR will improve the sensitivity, and will even enable operations during the night time. Even though we have corrected for dark signal variations with temperature and incorporated it in our current data analysis process, it may be important to look into details of the temperature dependence of the slit height and other optical parameters of the current

---

[4]To correct for the Ring effect which leads to the widening of spectral features in scattered light observations, e.g. de Beek et al., 2001



spectrograph. However, we are developing an in-house spectrograph with similar functionality but a better detector (see Sreejith et al., 2015), which we will be used in the future observations.

In these flights, our payload experienced frequent oscillations preventing us from continuous observation for a long duration. We have developed a pointing system (Nirmal et al., 2016), which will enable stable observations of regions of interest. We

will use the pointing system to carry out Multi-Axis DOAS (MAX-DOAS, Hönninger et al., 2004) observations using the same instrumentation set-up. In addition, we will be able to point at the Sun (or the Moon) to employ active DOAS technique of measuring trace gas strengths, thus improving the SNR and accuracy of trace gas analysis.

The same principle and the method used for the analysis of ozone can be applied to other trace gases in the atmosphere, such as BrO, $NO_2$, etc. We are planning to perform DOAS analysis of BrO. For this species, Aliwell et al. (2002), on the basis of

10 previous observations and DOAS analysis with different fitting windows, recommended to use the narrow wavelength band of 346–359 nm. They found that even a small change in the fitting window leads to significant changes in the retrieved SCDs, and therefore, the wavelength band of 346–359 nm was recommended.

*Acknowledgements.* Part of this research has been supported by the Department of Science and Technology (Government of India) under Grant IR/S2/PU-006/2012.



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

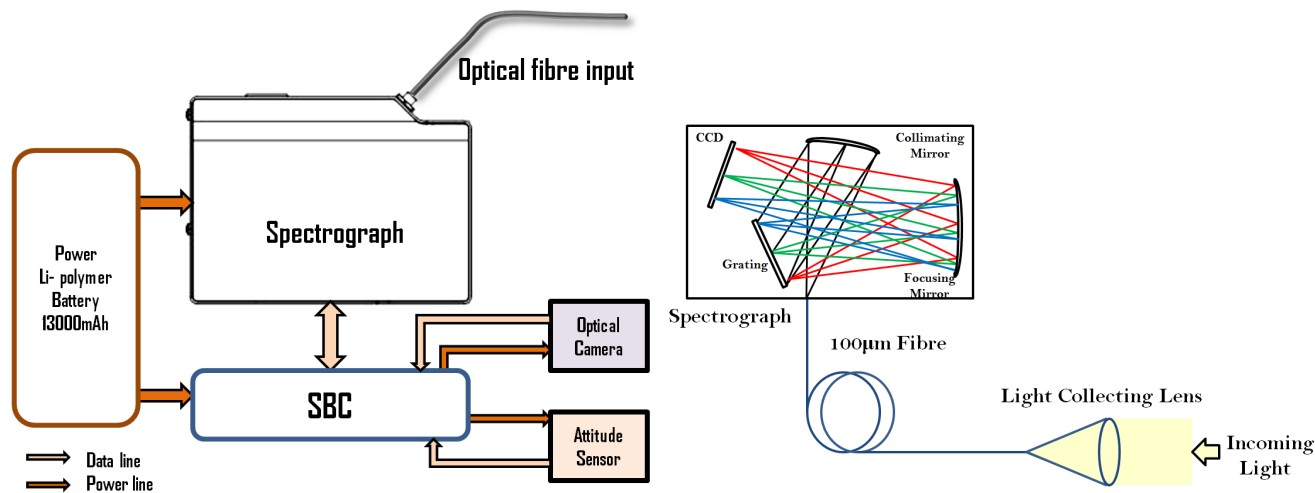

**Figure 1.** *Left*: Schematics of the instrumental set-up. *Right*: Schematics of the spectrograph.




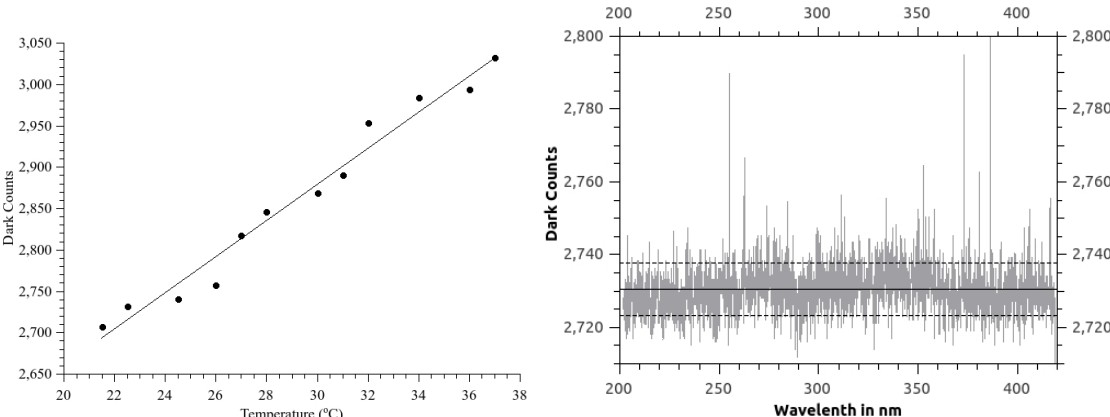

**Figure 2.** Results of the dark count calibration. *Left*: Dark counts/s vs. temperature. *Right*: Dark counts per second at 25°C. The solid line shows the mean value, and the dotted lines represent the standard deviation.

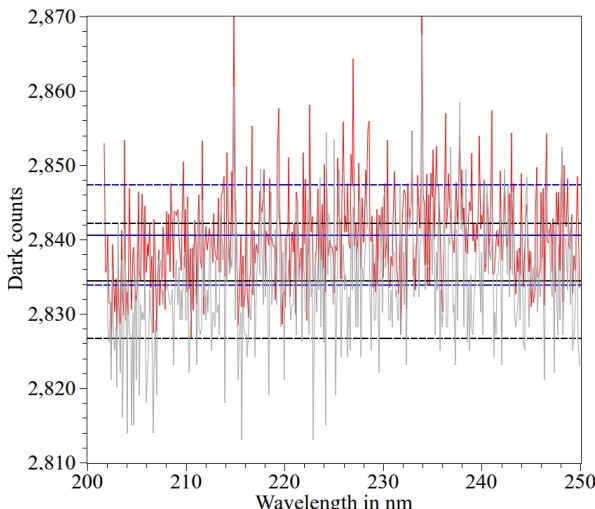

**Figure 3.** The variation of dark signal with wavelength in flight (grey) and in laboratory (red). The solid line represent the mean and the dotted line represent the standard deviation: black for flight data and blue for laboratory measurement at $28°C$.



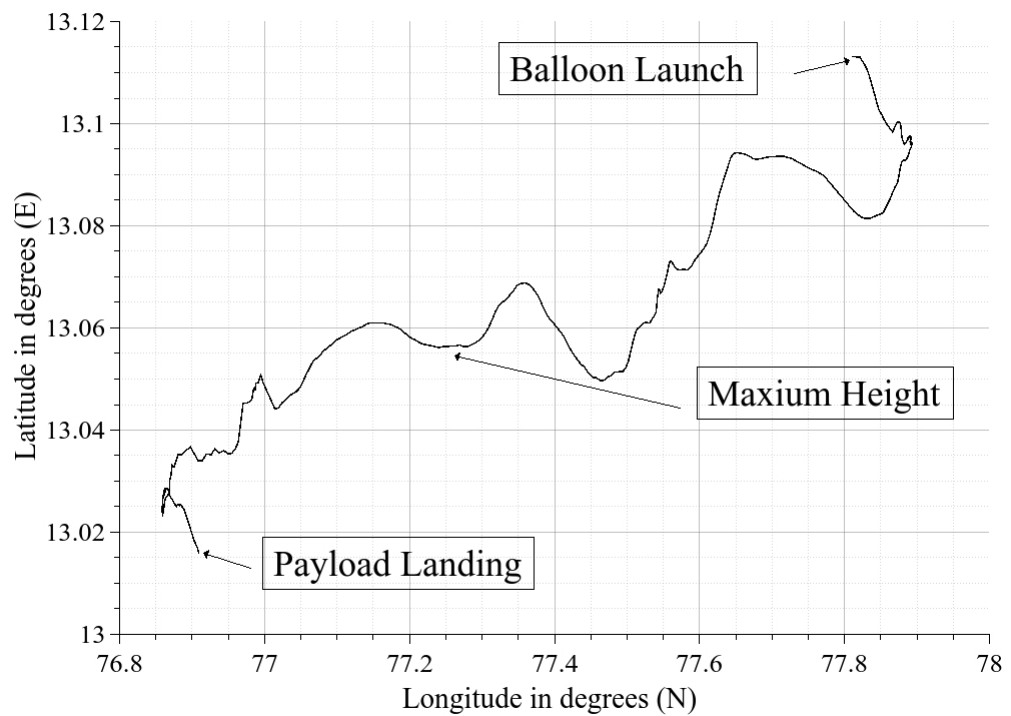

**Figure 4.** Balloon trajectory derived from GPS data for the June 15, 2014 flight.

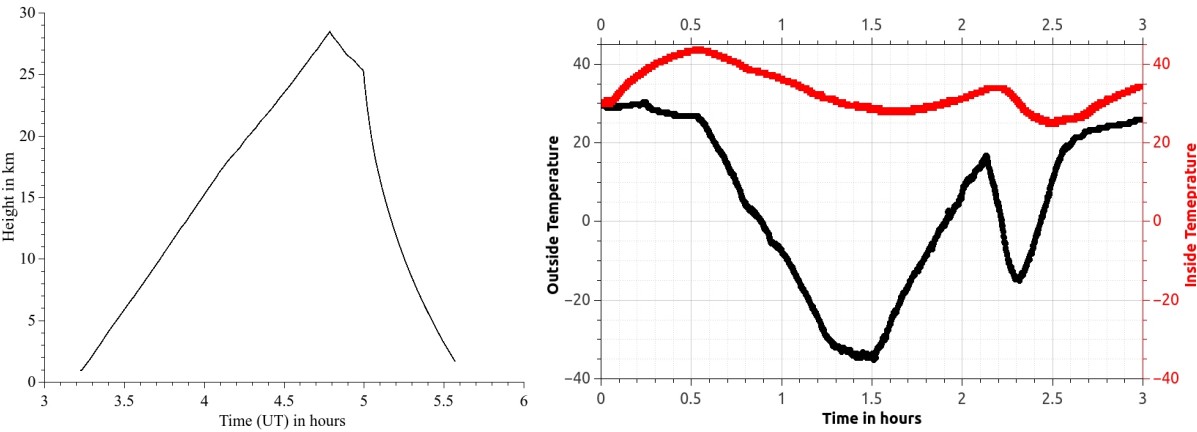

**Figure 5.** *Left*: Flight altitude profile on the June 15, 2014 flight. Outside temperature data was not available for this flight. *Right*: Variation of external (black curve) and internal (red curve) temperature during the October 12, 2014 flight. The sensors were switched on 30 minutes prior to the launch, therefore, the launch time corresponds to 0.5 hrs in the graph.




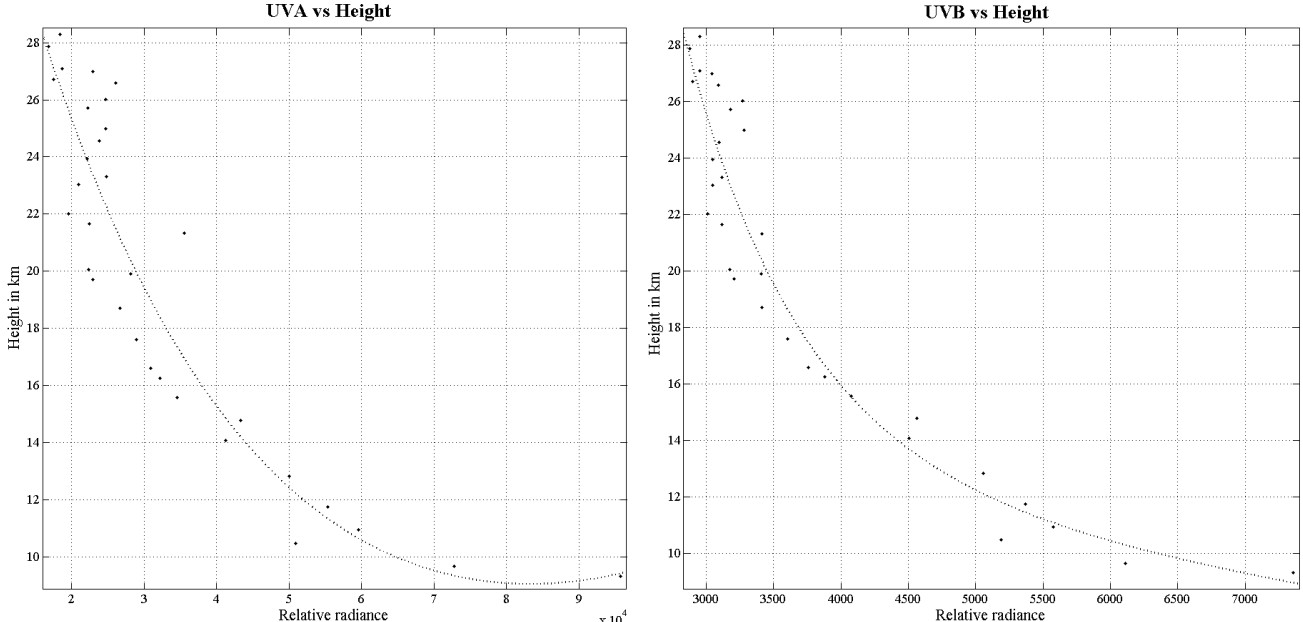

**Figure 6.** Variation of measured radiance with altitude. The lines represent exponential fit. Note the SZA change from 52 to 41 degrees with increase in altitude for June 15, 2014 flight.

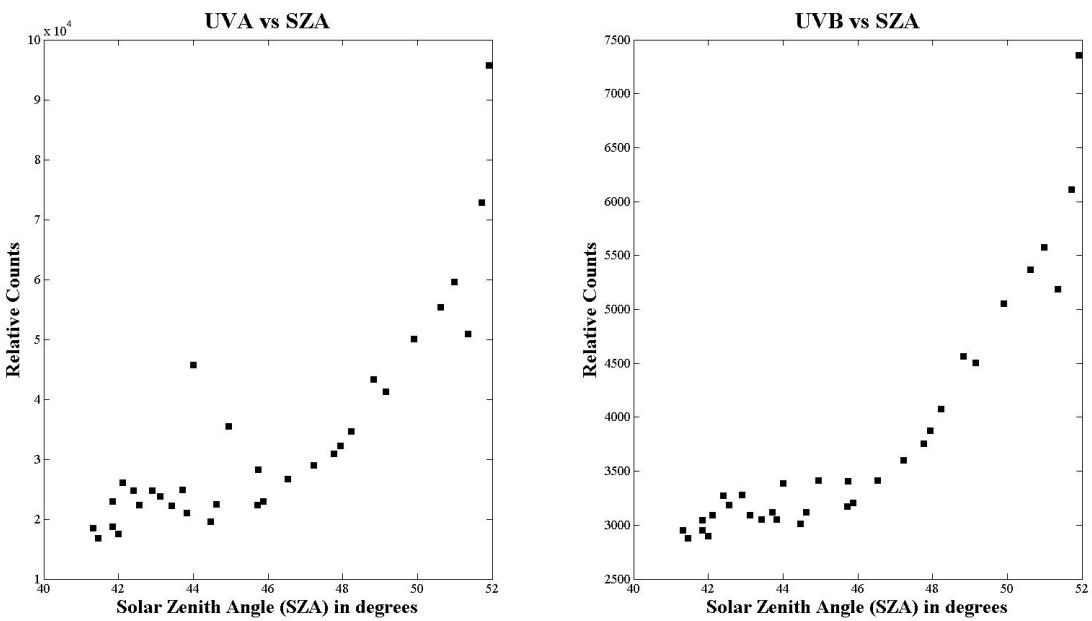

**Figure 7.** Variation of measured radiance with solar zenith angle on June 15, 2014 flight.



**Figure 8.** Sample DOAS analysis of ozone in wavelength band 305–340 nm. The figures show (clockwise from top): (a) the resultant fit of the data, (b) residual, (c) the fitted optical density of ozone and (d) synthetic ring spectrum. Red is the fit and blue corresponds to actual data in each case.



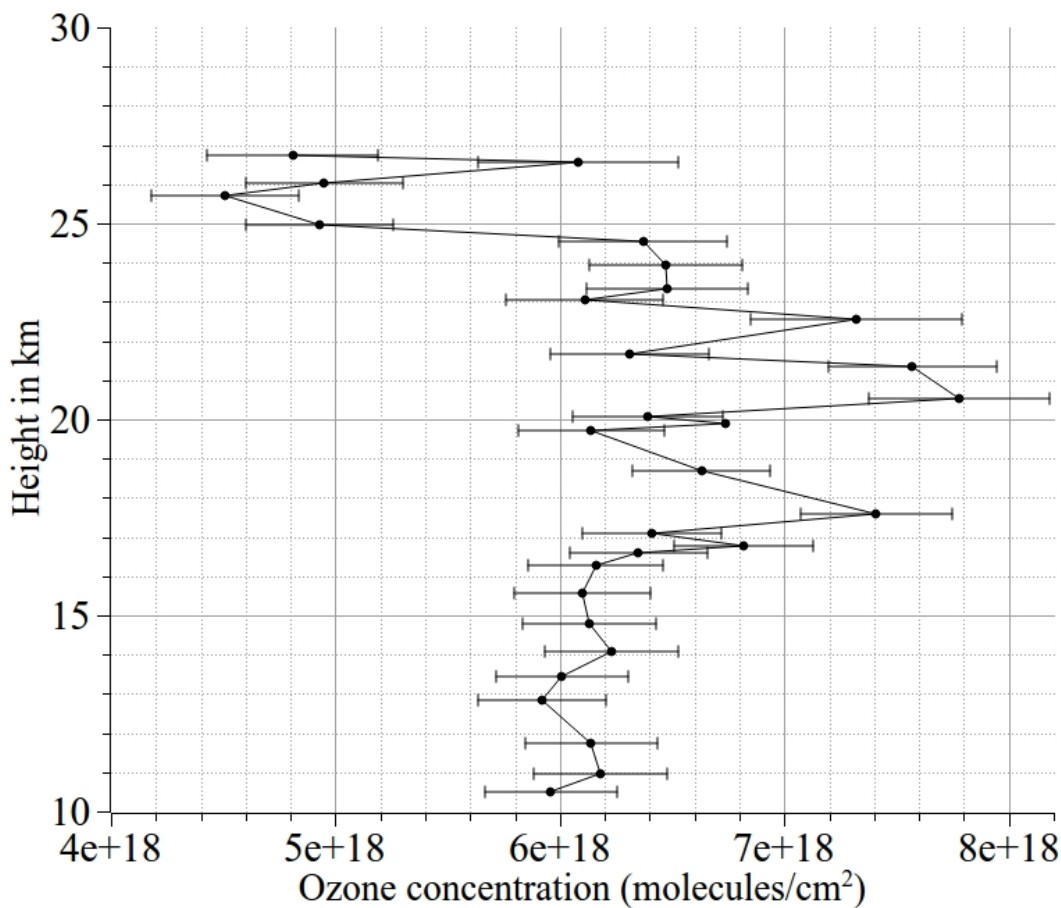

**Figure 9.** Observed ozone SCD on 15 June, 2014 flight.

**Table 1.** MayaPro2000 Technical Specifications

| | |
|---|---|
| Dimensions | $149 \times 109.3 \times 50.4$ mm |
| Weight | 0.96 kg (without power supply) |
| Power | 500 mA at +5V DC |
| Design | Symmetric crossed Czerny-Turner |
| Focal length (input) | F/4 101.6 mm |
| Signal-to-noise ratio | 45:1 (at integration time of 1 s) |
| Temperature | Operation: $0°C$ to $+50°C$ |
| Humidity | 0%–90% non-condensing |



**Table 2.** Attitude Sensor Technical Specifications

| | |
|---|---|
| Size | $86 \times 54 \times 45$ mm |
| Weight | $< 100$ g without battery |
| Power | 5 W |
| Components | Accelerometer, Gyroscope, Magnetometer and GPS |
| Accuracy | $0.48°$ (average RMS) |
| Output Modes | RA-DEC or Az-Ele |

**Table 3.** Environment sensors details

| Parameter | Sensor | Operating Range | Accuracy |
|---|---|---|---|
| Temperature | LM75 | $-55°$C to $+125°$C | $\pm2$K |
| Pressure | MSR145 FP/020 | 0 to 2000 mbar | $\pm2.5$ mbar |
| Humidity | MSR145 FH/020 | 0% to 100% | $\pm2$% |

**Table 4.** Details of the flights

| Date | Launch Site | Launch Time (local time) | Maximum Height | Touch Down (local time) | Remarks |
|---|---|---|---|---|---|
| 15.06.2014 | Hoskote, Bangalore | 08:15 | 28.7 km | 11:06 | Cloudy |
| 12.10.2014 | Hoskote, Bangalore | 07:00 | $\sim 25$ km | 09:45 | Partially cloudy |