# Peer review of "Measurement of limb radiance and Trace Gases in UV over Tropical region by Balloon-Borne Instruments - Flight Validation and Initial Results"

_Atmospheric Measurement Techniques, 2016_

## Referee Comment (RC1) · Anonymous Referee #1 · 6 Jun 2016

**Review for manuscript Atmos. Meas. Tech. Discuss., doi: 10.5194/amt-2016-98, 2016 by Sreejith et al.,**

The present paper reports on novel instrument for balloon-borne limb measurements of atmospheric radiance in the UVA, UVB, and UVC spectral ranges. From the measurements profiles atmospheric extinction and some atmospheric trace gases ($O_3$, $NO_2$, BrO, …) can be inferred using the well-known DOAS approach, together with radiative transfer modelling and mathematical inversion. Results from some first test flights are reported, but little on the post-flight data analysis. As such the paper is potentially worthy to become published in AMT, if it were to report on a consolidated instrument configuration and some final results together with validation exercise against any other measurements or model prediction. In this respect the manuscript does not really report on exciting novel achievements. Hence in the present form the manuscript may not really capture the interest of many AMT readers because it comes rather prematurely. Therefore I do not recommend the manuscript for publication.

Major comments:
(1) Section 2 starts with a description of the major features of the instrument, with some redundant information, c.f., for example the rate by which visual images were taken by an aboard camera, …. provided in section 3, while other information is largely missing. These include
   a. The field of view of the entrance telescopes
   b. The transmission of the instrument
   c. The slit size
   d. Information the spectrometer's f-number
   e. Optical stability with respect to spectrometer's temperature and ambient pressure, i.e. the T and P dependence of the parameters C1, C2, and 3 of the equation (1)
   f. The line shape as function of T, and p
   g. Sampling/pixel per full width half maximum as function of wavelength and line shapes
   h. Amount and wavelength dependence of the spectrometer stray-light
   i. Stability of the offset current with respect to T.
   j. Information how you calculated the signal to nose ratio (on page 6 denoted 2.06:1 by and in table 1 denoted by 45:1). For such an instrument, the SNR for single spectra could (and should) typically be several hundred, and for co-added spectra it could be some thousands or even better. In any case if true it is much too low for any type of reasonable scientific application.
   k. Also how is the chosen spectral interval justified (200 – 420nm), since in the limb the wavelength band (200 – 320 nm) may hardly provide any information on the targeted species.

(2) Measurements
   In order to assess the usefulness of the measurements, some information is missing.
   (a) Since the instrument (or the telescope) is not azimuth controlled, according to which criteria did you select the measured spectra for further analysis.
   (b) If true, the accuracy or stability of the elevation angle to within $\pm1°$ degree is rather coarse, and certainly too coarse to finally infer profile information on the targeted parameters. In order to assess the required elevation angle stability and FOV of the telescope (see point a. under (1)), information regarding the so-called averaging matrix need to be provided.
   (c) What is the sampling rate per altitude interval, and the targeted altitude resolution?
   (d) ….

(3) Analysis

(a) Equation 2) does not express a relative radiance, but rather a wavelength band integrated radiance. What are the wavelength limits of the integral?

(b) Equation 3 is rather poor mathematical representation of your measurements. How does the slit function, any broad band characteristics of the instrument, the spectrometer stray-light, et cetera… contribute to measured $I(\lambda)$.

(c) In Figure 6, what are the dotted lines and why the data scatter so much? Is the scatter telescope orientation dependent, and if yes how?

(d) Why should the (wavelength band integrated) relative counts (what is it? I guess it is a count rate) change with solar zenith angle. Explain and justify the result shown in Figure?

(e) How and to what accuracy the so-called Fraunhofer contribution is determined (i.e. amount of absorption in the reference spectrum).

(f) How can the spectral residual range between 12 order of magnitudes (Figure 8, panel b.), if the signal of noise ratio range at 45.1 at best?

(g) ….

---

## Referee Comment (RC2) · Anonymous Referee #2 · 14 Jun 2016

**General comments:**

This manuscript presents a newly developed balloon-borne UV spectrometer performing limb-scatter observations with the goal to retrieve (lower) stratospheric ozone concentration profiles. The instrumental setup is briefly described, two flights have been carried out to date, and some first results on the retrieval of O3 slant column densities are presented. The technical components appear to be relatively inexpensive and offthe-shelf, which makes this instrument attractive for other groups. In principle, I think a manuscript like that is of interest to the scientific community. The present manuscript,

however, lacks details and in-depth discussions typical of the usual AMT manuscript. The main section of the paper (4.2) covers barely one page, and the main result is an O3 slant column density profile. I ask the authors to retrieve a vertical density profile from this SCD profile. Otherwise, the reader cannot judge, how realistic the presented O3 SCD values are. My guess is that they are too low, but I may be wrong. In my opinion the manuscript requires a major revision, before it can become acceptable for publication in AMT.

Specific comments:

Title: "Flight Validation"

I don't fully understand the intended meaning of "flight validation". It suggests that some aspects of the balloon flights (trajectories ?) were validated, which was not the case, as far as I can tell. I suggest removing this from the title.

Page 1, line 7: "It can be used to measure the column densities"

Unclear what column densities – or what species – you refer to here.

Page 1, line 8: "upper atmosphere"

Please be more specific. For some people the upper atmosphere begins at the tropopause, for some at the mesopause.

Page 1, line 17: SOLSE/LORE is not a good example to highlight that spaceborne observations enable measurements over long periods of time, because SOLSE/LORE was shuttle based with a mission duration on the order of a week.

Page 2, line 8: "et al.," -> "et al."

Page 2, line 20: add space before "Right"

Page 3, equation 1: It would be good to list the values of the fit coefficients C\_x

Page 5, line 11: "The relative radiance in these bands is given by"
I don't understand why equation (2) corresponds to a "relative" radiance. I\_band is simply the mean radiance over the corresponding spectral range. Something is missing here.

Page 5, section 4.1:

The motivation for averaging over these two very wide spectral ranges (280 - 315 nm and 315 - 420 nm) remains unclear. Also, I'm not sure what the intended purpose of Figures 6 and 7 is. This should be made clear or the Figures should be removed.

Page 5, line 15: "the dotted lines represent an exponential fit in agreement with results obtained Weidner et al. (2005)"

It's not evident what aspect specifically is in agreement with the results by Weidner et al. The absolute values of the "relative" radiances?

```
Page 5, line 25: "broadband" -> "broad-band"
```

Section 4.2 (Trace atmospheric gases)

To me this should be the main section of the paper, demonstrating that this instrument is capable of providing robust observations of vertical ozone profiles. I think this section is far too superficial and lacks important details. Looking at Fig. 9, I would have expected slightly larger O3 SCDs. SCD values of  $6 \times 10+18$  / cm+2 roughly correspond to 220 DU, which is close to the vertical column for the location of the observations. It's difficult to tell without AMF calculations for the specific wavelength range used here, but I would expect the SCD to be larger than the vertical column density. Whether the retrieved SCD values are realistic can be checked either by retrieving a vertical O3 density profile or by using a RT-model to simulate the SCD profile for a realistic ozone density profile. In my opinion the first should be done before the paper is acceptable for publication in AMT.

Page 6, line 2: "trace gas strengths". "Strength" is not really a technical term, I think. Please use, e.g. slant column density.

**AMTD**
Page 6, line 22: "Our signal to noise ratio of 2.06"

Was the SNR really that low? Looking at panel a) and b) in Fig. 8 it seems to be much higher – at first glance.

Caption Figure 8: "(clockwise from top)". The Figures are not arranged clockwise.

Figure 9: The abscissa label says "ozone concentration", which is incorrect (ozone slant column density).

---

## Referee Comment (RC3) · Anonymous Referee #3 · 6 Jul 2016

The manuscript presents a technical set up to measure vertical distributions of atmospheric trace gases by a balloon-borne limb-viewing UV spectrometer. Although a technical setup is described sufficiently the manuscript does not report any significant scientific results. From the presented results it is unclear if the technique works reasonably well. Advantages and disadvantages of the technique with respect to the well-established ozone sonde measurements are not discussed. Similar measurements performed by other groups are not explicitly mentioned (although some of them are cited in an indirect way). It is absolutely insufficient to present ozone slant column density without converting them into the profiles as well as without presenting any comparison with independent data. Although it is difficult to judge from the profile of SCDs, my personal opinion is that the profile does not look as expected. Even in terms of SCDs I'd expect the maximum of ozone to be clearly pronounced. The paper does not discuss possible error sources, e.g. influence of pointing error or possible stray light. Unexpected features seen at the plots are not analyzed, these are e.g. an apparent harmonic oscillation seen in the dependence of the dark current on the temperature (Fig. 2), irregularities in the vertical behavior of the radiance with altitude (Fig. 6) or a funny peak seen in the UVA radiance at about 44 deg SZA which is not seen in UVB range (Fig. 7). The retrieval software seems to be used as a black box without thinking too much about the results coming out. For example, the ozone fit in Fig. 8c just cannot be true if a polynomial of the third order is subtracted as stated in the first paragraph at page 6, the fit for the ozone differential absorption structure in the 320 - 330 nm range often used for ozone retrievals looks just terrible. Furthermore, I doubt that the spectrum shown in Fig. 8d is really the ring spectrum (its source is not specified).

In general the paper is absolutely immature to be published in AMT. I'd like to encourage the authors to reconsider the paper by including a retrieval of ozone profile instead of SCDs, comparing to independent results and providing a comprehensive error analysis and then re-submit it.
* * *